# Quark Cheese Processed by Dense-Phase Carbon Dioxide: Shelf-Life Evaluation and Physiochemical, Rheological, Microstructural and Volatile Properties Assessment

**DOI:** 10.3390/foods11152340

**Published:** 2022-08-05

**Authors:** Xiaoyong Song, Yuanrong Zheng, Xuefu Zhou, Yun Deng

**Affiliations:** 1College of Energy and Power Engineering, North China University of Water Resources and Electric Power, Zhengzhou 450011, China; 2State Key Laboratory of Dairy Biotechnology, Shanghai Engineering Research Center of Dairy Biotechnology, Dairy Research Institute, Bright Dairy & Food Co. Ltd., Shanghai 201100, China; 3Department of Food Science & Technology, Shanghai Jiao Tong University, 800 Dongchuan Road, Shanghai 200240, China; 4Shanghai Food Safety and Engineering Technology Research Center, Shanghai 200240, China

**Keywords:** dense phase carbon dioxide, quark cheese, non-thermal processing, process optimization, food quality stability

## Abstract

Dense-phase carbon dioxide (DPCD), a novel non-thermal processing technology, has attracted extensive attention due to its excellent performance in food sterilization and enzyme inactivation without quality deterioration. In this work, we aimed to extend the shelf life of quark cheese with DPCD and explore the effect of DPCD treatment as well as storage time on the quality of quark cheese. The sterilization parameters were optimized by means of orthogonal experiments, and the physiochemical, rheological, microstructural and volatile properties of cheese were investigated. The optimal DPCD treatment (20 MPa, 45 min, 55 °C) successfully extended the shelf life of quark cheese due to its inhibition effect on yeast and was able to slow down the proteolysis and alterations in pH and color of cheese. Cheese processed using DPCD after 14-day storage even displayed similar rheological properties to the control at day 0, from which bound water significantly migrated during storage. Moreover, DPCD contributed to the retention of the volatile profile of cheese during storage. This study demonstrated that DPCD is a promising pasteurization technology for quark cheese to improve its quality stability during storage.

## 1. Introduction

Cheese, an increasingly widely consumed dairy product in recent years, is rich in nutrients. It possesses health-promoting benefits such as improving the metabolism and physical performance of the human body [1]. Quark cheese is a fresh unripened cheese prepared from pasteurized milk by means of acid or rennet coagulation which has many nutritional benefits and has been recommended as a part of a daily breakfast [1]. In addition, quark cheese has a mild, slightly sour flavor as well as a relatively light, soft and smooth texture, and thus it can serve as a flavor carrier and be used to enhance the textural profiles of products [2].

However, due to microbial contamination, quark cheese has a limited shelf life of about 2–3 weeks [2]. On the one hand, there is bacterial cross-contamination especially during and after curd production, which cannot be effectively controlled, despite the fact that the number of spoilage microorganisms in milk is substantially reduced through pasteurization [3]. Quark cheese, on the other hand, is very susceptible to microbial spoilage due to its high moisture content (around 82%) and high water activity, as this dairy product is an excellent substrate for the growth of spoilage microorganisms [4].

Thus, controlling the growth and reproduction of spoilage microorganisms is critical for extending shelf life. Traditional heat treatment has been used for cheese sterilization, but resulted in undesirable quality deterioration [5]. Non-thermal technologies, in comparison, have attracted a lot of attention in recent years due to their excellent sterilization and food quality retention [6].

Dense-phase carbon dioxide (DPCD), also known as high-pressure carbon dioxide (HPCD), is a promising non-thermal processing technology that can inactivate microorganisms and enzymes through the molecular effects of carbon dioxide (CO_2_) with pressurized CO_2_ below 50 MPa and a treatment temperature below 60 °C [6,7]. Its good applicability as well as its capacity to inactivate microorganisms and enzymes without serious quality deterioration mean that DPCD processing technology has been widely investigated in liquid foods [8], solid foods [7] and even solid–liquid food mixtures [6]. Nowadays, the novel dairy processing method has been widely explored. However, considering the pH-lowering effect of DPCD, this technology appears to be unsuitable for application in foods susceptible to acidity, such as milk. As reported by Liao et al. [9], an aggregation of casein could be caused by DPCD due to a decrease in pH. Therefore, cheese seems to be a great substrate for DPCD treatment due to its acidic matrix, and quark cheese seems to be more easily fully penetrated by CO_2_ to exert the molecular effect because of its soft and semi-solid texture [2]. As far as we know, there is currently no research on the application of DPCD in quark cheese processing. Additionally, it is promising that the combination of microbiological and enzymatic inactivation caused by DPCD and the CO_2_-modified atmosphere during cheese storage could better control the microorganisms and retain the product quality [10].

In this work, we utilized DPCD processing technology in quark cheese sterilization. The DPCD parameters were optimized and the effect of DPCD on microbial, color, proteolytic, microstructural, rheological and volatile properties were investigated. This study will provide a basis for further utilization of DPCD in dairy products.

## 2. Materials and Methods

### 2.1. Materials

Commercial pasteurized (75 °C for 10~15 s) skimmed cow’s milk was purchased from a local supermarket (Shanghai, China) and was stored at 4 °C. The commercial rennet as well as a starter culture including *Lactococcus lactis* ssp. *lactis* and *Lactococcus lactis* ssp. *cremoris* were bought from Doit Biological Technology (Beijing, China).

### 2.2. DPCD Parameter Optimization

#### 2.2.1. Preparation of Quark Cheese

A volume of 350 mL milk pre-heated at 31 °C for 30 min was inoculated with 4.8 mg of the starter culture. After 30 min of pre-fermentation at 31 °C, 0.2 mg of rennet was added to the mixture. The milk was then incubated at 31 °C for 10.5 h for fermentation. Finally, the curd was moved to soft gauze bag and allowed to stand for 8 h to remove whey at 12 °C. The quark cheese samples were stored in sterilized bags at 4 °C until use.

#### 2.2.2. DPCD Treatment

Treatments of DPCD were performed in a batch apparatus (HKY-1, Hai’an Petroleum Scientific Research Instrument Co., Jiangsu, China) as described by Tang et al. [6]. For each experiment, 30 g quark cheese samples were placed in a high-pressure vessel. After the vessel was preheated to the set temperature, it was then pressurized with CO_2_ until the pressure reached the experimental setting. The cheese samples were held at a constant pressure at an equilibrium temperature during DPCD treatment, and then the vessel was slowly depressurized. Quark cheese samples were collected and stored in sterilized sealing bags at 4 °C after DPCD treatment.

#### 2.2.3. Orthogonal Optimization

Referring to the suitable range of DPCD treatment parameters reported by Tang et al. [6], Liao et al. [9] and Guo et al. [11], as well as our preliminary experimental results, an orthogonal experiment design L_9_ (3^3^) (Table 1) was used to determine the optimal DPCD treatment parameters for quark cheese sterilization based on the total bacteria count (TBC).

#### 2.2.4. Microbiological Analyses

Total bacterial count (TBC) and total yeast and mold count (TYMC) were assessed according to Tang et al. [6] and Seyed-Moslemi et al. [1] with slight modifications. Cheese samples (1 g) were homogenized with 10 mL of NaCl solution (0.85%). The mixture was then serially diluted with NaCl solution. Each dilution (1 mL) was added to plates containing plate count agar (PCA) for total bacterial count (TBC) measurement and potato dextrose agar (PDA) for total yeast and mold count (TYMC) measurement. Observation and counting should be conducted after 2 days of cultivation at 37 °C for TBC and 5 days at 28 °C for TYMC.

### 2.3. Analysis of Quark Cheese Properties during Storage

To further evaluate the effect of DPCD on the quality change of cheese samples during storage, the cheese was then treated with DPCD with optimized parameters and was stored at 4 °C. The quark cheese without DPCD treatment was labeled as the control.

#### 2.3.1. Microbiological Analyses

The microbial analyses during storage were conducted on day 0, 7, and 14. See Section 2.2.4. for more detailed information.

#### 2.3.2. Sensory Evaluation

Sensory evaluation was conducted referring to the method reported by Tang et al. [6]. The sensory index grades of the cheese samples were classified as qualified and unqualified. Qualified quark cheese requires the following characteristics: it should be milky white or yellowish; it should have a strong milk flavor, slightly sour without a bitter taste; it should be soft, elastic, fine, smooth and easy to apply and have a light sense of graininess. Additionally, cheese without the above characteristics is considered unqualified. With reference to the above evaluation criteria, 7 semi-trained panelists were involved in the judgement on whether the cheese samples were qualified. The results depended on the majority.

#### 2.3.3. Color

Changes in color parameters including the L*, a*, and b* values of cheese samples were determined using a LabScan XE color difference meter (HunterLab, Reston, VA, USA). This was conducted on day 0, 7, and 14 during cheese storage. The L*, a* and b* values describe the visual lightness, redness and yellowness of the cheese samples, respectively.

#### 2.3.4. Proteolytic Activity

The proteolytic activity of quark cheese was estimated through the reaction between o-phthaldialdehyde (OPA) and free amino acid groups as previously reported by Silva et al. [10]. First, the mixture containing 3 g of cheese and 27 mL of water was centrifuged at 5000 rpm for 10 min at 10 °C (Z-326 K, Hermle Labortechnik GmbH, German), and 9 mL of supernatant was then mixed with 3 mL of 40% (*w*/*v*) trichloroacetic acid solution (TCA). After that, this mixture was centrifuged at 5000 rpm for 15 min at 4 °C and 0.15 mL of supernatant was added to 3 mL of the OPA reagent solution. Finally, the absorbance of the mixture at 340 nm was determined and reflected the relative proteolytic activity of the cheese. It was measured on day 0, 7, and 14 during cheese storage.

#### 2.3.5. Microstructure

The microstructure of cheese samples was investigated using the confocal scanning laser microscopy (CSLM) method as previously described with slight modifications [12]. Cheese slices ~1 mm thick were stained with 200 mg/L fluorescein isothiocyanate (FITC) acetone solution for 1 min and 200 mg/L Nile red acetone solution for 1 min, respectively. The samples were then placed on a cover glass and observed using an IX71 FV300 confocal laser scanning microscope (Olympus Optical Co., Tokyo, Japan) in fluorescence mode. Filters of the wavelengths 513 nm and 633 nm were used to capture the fluorescence of FITC and Nile red, respectively. This microstructural observation was conducted on day 0 and 14 during cheese storage.

#### 2.3.6. Rheological Properties

The rheological properties of cheese samples were measured using a Haake RS 6000 rheometer (Thermo Electron, Langenselbold, Germany). Before the measurement, cheese samples were stabilized at room temperature for 30 min. The storage modulus (*G*′, Pa) and loss modulus (*G*′, Pa) were measured in an oscillatory frequency sweep mode with an invariable strain of 1.0% and a frequency range of 0–10 Hz. *k*′ and *k*″ values were calculated according to the power law models. The formulas were as follows:$$G^{\prime }=k^{\prime }\cdot \omega^{n^{\prime }}$$
$$G^{\prime }=k^{\prime \prime }\cdot \omega^{n^{\prime \prime }}$$
where *ω* is the angular frequency, and *n*′ and *n*″ are the frequency exponent.

The steady shear behavior of cheese samples was determined by measuring the apparent viscosity of cheeses as a function of shear rate from 0.1 to 100 s^−1^, at a constant frequency of 1 Hz and a constant temperature of 25 °C. The *n* and *k* values were calculated according to the following formula:$$\eta =k\cdot \gamma^{\left(n-1\right)}$$
where *η* is the apparent viscosity, *γ* is the shear rate, k is the consistency index and *n* is the flow behavior index. The measurement of rheological properties was conducted on day 0 and 14 during cheese storage.

#### 2.3.7. Moisture Distribution

A low-field ^1^H NMR (21 MHz, Niumag Electronics Technology, Shanghai, China) was used to evaluate the change in water mobility and distribution of cheese samples during storage. Approximately 2 g of the cheese sample was placed in the center of the radio frequency coil at the center of the permanent magnetic field and used for CPMG scanning experiments. The typical pulse parameters were as follows: sampling frequency (SW) = 100 kHz, offset frequency (O) = 237,734.33 Hz, time echo (TE) = 0.5 ms, pulse time of 90° (P1) = 10 μs, pulse time of 180° (P2) = 34.04 μs, time waiting (TW) = 5000 ms, number of sampling points (TD) = 75,002, number of scans (NS) = 8, and number of echoes (NECH) = 1500.

As the absolute relaxation amplitudes were proportional to the amount of water present, the relative amplitudes within samples were used. The signal per mass of the magnitude parameter of the *ith* exponential was used to reflect the ability of samples to absorb the corresponding water and was calculated as follows:$$Signal\;per\;mass\left(au\cdot ms\cdot g^{-1}\right)=\frac{A_{2i}}{m}$$
where *A*_2*i*_ is the corresponding water population (area ratio) of the *ith* component and m is the sample mass. The signal per mass of total water is the sum of each signal per mass component. This measurement was conducted on day 0 and 14 during cheese storage.

#### 2.3.8. Volatile Profile

Gas chromatography-mass spectrometry (GC-MS) was conducted using head space solid-phase microextraction at 50 °C for 30 min with a polydimethylsiloxane/divinylbenzene (65 μm) extraction head and a 5 min desorption time. GC-MS detection utilized an elastic quartz capillary DB-WAX column (30 m × 0.25 mm × 0.25 μm, Agilent, Santa Clara, CA, USA) and a gas chromatograph-mass spectrometer (Agilent, Santa Clara, CA, USA) under the following conditions: 40 °C for 5 min, rising to 220 °C at 5 °C/min and to 250 °C at 20 °C/min and these conditions were maintained for 2.5 min.

This measurement was conducted on day 0 and 14 during cheese storage. The data were analyzed using MSD chemical analysis software, and quantitative and qualitative analysis of the data was carried out with the NIST 14.L spectral database (https://www.nist.gov, accessed on 13 September 2021).

### 2.4. Statistical Analysis

Experimental data were analyzed by ANOVA, and significant differences at *p* < 0.05 were determined using Duncan’s multiple range tests in the SPSS 21.0 statistical data analytical software (IBM, New York City, NY, USA). SIMCA 14.1 (Umetrics, Umea, Sweden) was used to further explore the similarities and differences in volatile components. The multivariate statistical analysis included unsupervised principal component analysis (PCA) and supervised orthogonal partial least squares discriminant analysis (OPLS-DA).

## 3. Results and Discussions

### 3.1. Orthogonal Optimization of DPCD Parameters

TBC was significantly reduced by DPCD treatment, and the germicidal efficacy was closely related to the DPCD parameters shown in Table 1. Temperature was the most important factor, followed by time and pressure, according to the range analysis results (Table 1), since R_C_ > R_B_ > R_A_. The variance analysis results (Table 2) indicate that temperature and treatment time had a significant impact (*p* < 0.05) on the bacteria inactivation in cheese samples. Lactic acid bacteria (LAB) acted as the major microorganisms in our cheese samples, which were sensitive to temperature rise and the prolonging of time when treated by DPCD [13]. Although an increase in temperature reduced the solubility of CO_2_, it also increased CO_2_ diffusivity and cell membrane fluidity, resulting in cell component extraction, protein denaturation and cytoplasmic membrane collapse [14]. The importance of temperature in DPCD treatment was also demonstrated by Ji et al. [14] and Tang et al. [6]. Pressure was found to play the least important role in microbial inactivation of quark cheese samples. This was consistent with the contribution of pressure to sterilization in paprika with DPCD [15]. On the contrary, the significant effects of DPCD pressure on microbial reduction have been reported elsewhere, such as in coconut water [16]. This varying effect of pressure could be due to material statuses, with solid foods being less sensitive to pressure variation in terms of microbial inactivation [6].

The range analysis results exhibited that 30 MPa (K_1_ > K_2_ > K_3_), 45 min (K_2_ > K_1_ > K_3_) and 55 °C (K_1_ > K_2_ > K_3_) were optimal parameters for DPCD treatment. However, according to variance analysis for the pressure term (data not shown), there was no significant difference between the 30 MPa and 20 MPa groups. Higher pressure would cause higher costs and severe protein degeneration [11], so 20 MPa was the optimal choice. Thus 20 MPa, 45 min and 55 °C (A_2_B_3_C_3_) were considered as the optimal DPCD treatment parameters for bacteria reduction.

### 3.2. Shelf-Life Evaluation

Table 3 displayed the changes in the quantity of microorganisms in cheese samples during storage. The TBC in untreated cheese was up to 9.77 Log CFU/g and was composed primarily of LAB [13]. After DPCD processing under optimal conditions, the TBC in cheese decreased by nearly seven orders of magnitude before storage. It was observed that the level of TBC in cheese processed by DPCD remained low throughout the storage. This might indicate a slower microbial metabolism, degradation of proteins and lipids, and catabolism of amino acids and fatty acids, all of which produced numerous compounds that affect the flavor and texture of cheese [17]. In terms of TYMC, there was initially no mold or yeast detected in our cheese samples, probably due to the strict hygienic conditions implemented by the milk company and our laboratory. After 14 days of storage, mold and yeast began to be detected in both cheese samples, which were all identified as yeast based on colony morphology. We found that TYMC in the control was 66.50 CFU/g after 14 days, which is considered to be excessive and was unsuitable for further storage. However, cheese processed by DPCD was still microbially safe after 14 days, as the yeast count was significantly less than 50 CFU/g at 3.50 CFU/g. This was mainly attributed to the presence of CO_2_ in the storage environment of cheese that inhibited the growth of spoilage microorganisms [10]. Apart from microbial factors, the sensory evaluation also determines the shelf life of cheese. It could be observed that both cheeses were all deemed to be sensorially qualified within the microbially safe period. Thus, in this work, DPCD significantly extended the shelf life of quark cheese by inhibiting yeast growth during storage.

### 3.3. pH

It is shown in Table 3 that DPCD processing resulted in a slight increase in the pH value of quark cheese at day 0 (*p* < 0.05). This pH increase in cheese could also be achieved by high-pressure processing [18]. However, according to Liao et al. [19], the pH decreased obviously after DPCD treatment. The pH value and the state of food matrix itself were responsible for such difference in pH alterations. For our quark cheese samples, dissolution of CO_2_ in this semi-solid matrix were limited, which restricted the dissociation of hydrogen ions. However, the dissociation of ionizable groups (e.g., carboxyl groups) in protein systems caused by DPCD might have played an important role in lowering the pH of cheese [18]. During the storage, there was a significant increase in pH value as time progressed. This could be attributed to the metabolic decomposition of lactic acid by yeasts or non-starter bacteria [20]. Additionally, after 14 days of storage, pH rose from 4.18 to 4.42 in the control, slightly more than the DPCD-processed cheese. It was the DPCD-inactivated enzymes and fewer microorganisms that contributed to the smaller pH alteration [6,20].

### 3.4. Proteolytic Analysis

During storage, proteolysis is inevitable in cheese due to enzymatic decomposition during storage. In this study, the proteolytic activity of quark cheese ranged from 0.287% to 0.348%. (Table 3). The minimal effect of DPCD processing on the proteolytic activity of cheese on day 0 suggested that the molecular effect of CO_2_ under this pressure and temperature did not result in protein degradation. This was consistent with the results reported by Liao et al. [9], who found that the peptide chain of lipoxygenase subunits could not be decomposed by DPCD treatment. For all cheese samples, there was a significant increase in proteolytic activity during storage. Additionally, we could see that the proteolytic activity index of the control increased from 0.0296 to 0.0348, which was more intense than the change in the DPCD-processed cheese. This demonstrated a slower proteolytic action in cheese with DPCD treatment during storage. On the one hand, proteases such as plasmin and rennet in cheese samples could possibly be inactivated by structural alterations [11,19], which consequently preserved milk proteins from vigorous hydrolysis. On the other hand, smaller amounts of microorganisms in DPCD-processed cheese, as mentioned previously, might lead to less involvement of proteins in the metabolism of LAB and yeast. Silva et al. [10] also found that the presence of CO_2_, effective in controlling spoilage, was associated with lower proteolysis in cheese. Similar inhibition of proteolysis occurred in cheese processed using high-pressure treatment [21,22]. Although the generation of polypeptides and amino acids caused by proteolysis contributes to the unique flavor of cheese, it is also responsible for undesirable flavors such as maltiness and bitterness [3,23].

### 3.5. Color Changes

Color is known as an important factor for cheese quality, and the effect of DPCD treatment on cheese color is shown in Table 4. After DPCD treatment at day 0, the L* value of cheese hardly changed, while the a* value decreased from −0.47 to −1.04 and the b* value increased from 10.32 to 11.05 significantly (*p* < 0.05), indicating greener and yellower cheeses. This was consistent with the color change of cheese after high-pressure treatment reported by Evert-Arriagada et al. [3]. It is important to highlight that DPCD treatment exhibited greater advantages in preserving cheese color than traditional heat sterilization according to the fact that treatment at 117 °C for 20 min would inevitably cause the cheese color to degrade visually [5], while changes in cheese color were indistinguishable following DPCD treatment in this work referring to sensory descriptions. During the storage, a significant increase in L* value could be observed in the control ranging from 94.13 to 94.84 (*p* < 0.05), which was greater than the DPCD-processed cheese. This was because protein hydration decreased in quark cheese samples and the increased number of free moisture droplets consequently increased the degree of light scattering [18]. Additionally, we found an obvious increase in redness and decrease in yellowness in our cheese samples, which are possibly related to changes in the cheese structure such as the protein network during storage [24].

### 3.6. Microstructure

The microstructure of cheese samples was monitored using CSLM to determine the changes in the protein matrix (stained in green) and fat distributions (stained in red). It could be observed that a relatively continuous protein matrix existed in the control initially, accompanied, however, by slight heterogeneity and a dispersed fat phase in the form of discrete and large globules (Figure 1). This could be explained by the milk used for the cheese preparation not being homogenized, which probably resulted in an increase in particles and a lack of active fillers (homogenized milk fat globules) in the acidified casein network [25]. At day 0, an ununiform and loose microstructure of cheese resulted from DPCD treatment distinguishably, since many black cracks appeared in the protein matrix. This was due to the fact that DPCD treatment could lead to aggregation of milk proteins in cheese by enhancing the protein–protein interactions [11,19]. The transition from its protein hydration to protein–protein interactions then resulted in a more discontinuous protein matrix [26]. Additionally, this supported our sensory descriptions stating that DPCD-processed cheese had a slightly gritty flavor. After the storage for 14 days, we found an obvious extension of the serum phase and a concomitant reduction in the size of, and an increase in the number of, fat globules in the control. Similar microstructural changes have also been described in cheese subjected to high-pressure treatment [27]. However, there was no distinguishable change in the microstructure of DPCD-processed cheese. Combined with proteolytic analysis, the intense proteolysis might lead to a weakening of the connection between casein as well as a loosening of the protein skeleton structure, which could explain the greater extension of the serum phase in the control.

### 3.7. Rheological Properties

Processing with DPCD and the storage times affected the rheological performance of cheese samples, as displayed in Figure 2. The storage modulus (*G*′) reflects the ability of cheese to store energy while maintaining its complete structure and loss modulus (*G*″) indicating the ability to dissipate mechanical energy by converting mechanical energy into heat through molecular motion [28]. For all cheeses, we found that *G*′ > *G*″, demonstrating the elastic structure of our quark cheeses (Figure 2A,B). This was consistent with the strain sweep test results for Akawi cheese reported by Abdalla et al. [29]. The viscoelastic properties (*G*′ and *G*″) of cheese were obviously enhanced by DPCD treatment. However, these values were diminished during storage for both cheeses. Power law rheological parameters (*k*′, *n*′, *k*′, *n*″) were calculated and are shown in Appendix A to further characterize the viscoelastic properties of cheese. The values of *n*′ and *n*″ reflected the frequency dependence of rheological parameters and were used to describe the type of bonding of the structural elements present in the matrix [30]. These values for all cheese samples were always lower than 0.20, which is considered a relatively low level, indicating the presence of strong and cross-linked gels with permanent covalent bonds [30]. We found a significant increase in *n*′ value following DPCD treatment on day 0, indicating an increase in the frequency dependence of rheological moduli. This suggested a less structured cheese matrix resulted from DPCD processing. As we previously discussed, this was consistent with the more discontinuous protein phase and extended serum phase of cheese after DPCD treatment.

Shear properties of cheese should be estimated in the complete characterization of rheological profiles. The change in apparent viscosity with shear rate is exhibited in Figure 2C. It could be observed that the apparent viscosity of cheese decreased continuously with increasing shear rate, indicating the shear thinning flow behavior. The flow behavior index (*n*) and the consistency index (*k*) were also calculated through the power law model and displayed in Appendix A. Similarly to the viscoelastic results, the cheese processed with DPCD showed lower *n* values than the control at day 0. This demonstrated the weakened ability to resist shearing led by DPCD because of the discontinuous protein network structure. However, DPCD significantly increased the apparent viscosity of cheese at day 0 according to Figure 2C and *k* value changes in Appendix A. Changes in moisture distribution, which has been confirmed to be closely related to the rheological properties of cheese, were most likely to blame [30]. The prolonging of storage time caused an obvious decrease in apparent viscosity, which corresponded to alterations in *k*′ and *k*″ values. The proteolysis might be responsible for this phenomenon. Surprisingly, DPCD-processed cheese after 14 days of storage displayed similar rheological properties to the control at day 0. So, DPCD treatment could somewhat compensate for the degradation of cheese’s rheological properties during storage.

### 3.8. Moisture Distribution

The proton relaxation signal of the cheeses was investigated, which was affected by the cheese matrix due to interactions between water and macromolecules [31]. Due to the heterogeneity of our cheese samples, five different proton populations were identified (Table 5). The water population peaked at around 0.15 ms, which could be attributed to the ^1^H of water being strongly bound to the casein structure [30]. DPCD treatment resulted in a slight increase in T_21_ at day 0 according to Table 5. This might be due to the enhancement of casein surface hydrophobicity caused by DPCD, which consequently weakened the protein hydration [26]. This was consistent with the slight extension of the serum phase in cheese microstructure resulting from DPCD. The water in T_23_ and T_24_ was attributed to protons of water trapped in the protein meshes and considered as immobilized-state water [30,31]. We also found that the DPCD-processed cheese initially displayed significantly lower T_23_ and T_24_ on day 0, which was consistent with the rheological properties. Tidona et al. [30] also illustrated a similar negative correlation between the relaxation times of these two populations and viscoelasticity. After 14 days of storage, it could be observed that T_21_ decreased significantly and the population in T_22_ migrated and became undetectable in the control (*p* < 0.05). This was probably due to the intense proteolysis leading to an alteration in protein structure as well as a destruction of the protein network [32].

### 3.9. Volatile Profile

Flavor is recognized as a critical factor in determining consumer acceptance of cheese. The flavor of cheese is produced by a combination of a large number of volatile substances, which are mostly derived from complex biochemical reactions during the preparation and deterioration of cheese [33]. A total of 40 volatile compounds were detected in our cheese samples including sulphocompounds and aliphatic hydrocarbons, heterocyclic compounds, acids, alcohols, aldehydes, ketones, ethers, and aromatic hydrocarbons, as shown in Table 6. There was significant reduction (*p* < 0.05) in the contents of acids including acetic acid, butyric acid, caproic acid, caprylic acid, and *n*-decanoic acid in cheese after treatment with DPCD, among which the acetic acid content decreased most obviously (Appendix A). Similar results have also been carried out by Evert-Arriagada et al. [34], who found that the content of acids in cheese volatile compounds was significantly reduced after high-pressure treatment. There was a relatively high content of alcohols in cheeses, and this could be possibly explained by the use of a high incubation temperature, which might not correspond to the mesophilic starters used in the study. Evert-Arriagada et al. [34] also reported such initial alcohol content in alcohols. Significant alterations in the volatiles of cheese were determined after 14 days of storage. Due to its lower odor threshold, sulphocompounds with a strong odor of garlic, onion, cabbage and mature cheese had an important contribution to the flavor of cheese, though the detected content is very low [35,36]. However, dimethyl disulfide is the only sulphocompound detected in cheese, and it became undetectable after 14 days of storage for both cheese samples.

Before the storage, three aliphatic hydrocarbons were detected in our cheese samples, among which 2,4-dimethylheptane was the main component. After 14 days of storage, there was no significant change in the content of aliphatic hydrocarbons, but a wider variety of aliphatic hydrocarbons became detectable, including decane, undecane, and d-limonene, and octane occupies the greatest content in aliphatic hydrocarbons. The formation of new aliphatic hydrocarbons during storage might be attributed to the oxidation of fats [37]. However, aliphatic hydrocarbons were not the critical component for cheese flavor due to their high odor threshold [38]. In terms of acids, their content decreased after storage in the control, which was mainly attributed to the decrease in acetic acid content. This could be related to the role of acetic acid as an intermediate in biochemical pathways [39]. Such transformation was inhibited by DPCD treatment, as the acetic acid content remained constant in DPCD-processed cheese, which could possibly be explained by the weak metabolism. Similar results were obtained by Lues and Bekker [40] and Upreti et al. [41], who found that the content of acetic acid dropped rapidly during cheese storage. Additionally, butanoic acid and caproic acid increased obviously in both cheeses after storage, which played dominant role in total acid content alterations of DPCD processed cheese. Butanoic acid was derived from the fermentation of lactose and lactic acid, which changed greatly in the control with the existence of a large amount of yeast [41]. In terms of aldehydes, the quantity and content of aldehydes detected were low. Additionally, these compounds were mainly derived from amino acids either by transamination, leading to an intermediate imide that can be decarboxylated, or by Strecker degradation [42]. Aldehydes are short-lived compounds in cheese, which can be rapidly reduced to primary alcohols and even oxidized to corresponding acids. This is the reason why there was only a small number of aldehydes with low content detected. However, it has been confirmed that aldehydes, especially linear aldehydes, have an important contribution to the freshness and floral aroma of cheese [42]. The contents of hexanal and nonanal increased significantly in DPCD-processed cheese after storage, while they remained unchanged in the control. Therefore, DPCD treatment might have the potential to maintain the freshness of cheese flavor. Additionally, acetoin is a critical compound of cheese that contributes to the desired creamy aroma [42]. Due to the proteolysis during storage, the acetoin content in both cheeses significantly increased [23]. *Lactococcus lactis* ssp. *cremoris*, more vulnerable to heat and high pressure than *Lactococcus lactis* ssp. *lactis*, was considered more suitable for flavor development and contributed to acetoin generation. This could explain the limited formation of acetoin in DPCD processed cheese after storage [43,44,45].

Our GC-MS results are equivocal in demonstrating the clear differences between the volatile profiles of cheese with or without DPCD treatment. Therefore, principal component analysis (PCA) was used to reduce the dimensions of our GC-MS data. Four principal components were extracted with a characteristic value > 1. PC1 and PC2 possessed variances of 43.9% and 22.9% (Appendix A), respectively, totaling 66.8%. According to the PCA score plots, the control day 0 group showed high similarity with the DPCD treatment group, indicating that the flavor property of cheese was hardly affected by DPCD treatment before storage overall. The control day 14 group and the DPCD treatment day 14 group were obviously isolated from the groups at day 0, and they could be clearly distinguished as they were located in the second and third quadrants, respectively. Therefore, DPCD treatment resulted in different changes in the cheese volatile profile during storage. The changes in *Lactococcus lactis* ssp. *lactis* and *Lactococcus lactis* ssp. *cremoris* by DPCD that led to changes in casein breakdown and flavor production from amino acids were considered one of the reasons for this difference [45].

The OPLS-DA model was also used to establish the volatile components that led to the PCA score difference between the control day 14 group and the DPCD treatment day 14 group. The R^2^Y and Q^2^ values for the model were 0.978 and 0.933, respectively, and indicated a good predictive ability. The VIP values for these two classes are shown in Appendix A. Components with VIP values ≥1 are marked in red, and are considered as the differential metabolites of the two groups [46]. Ethanol, acetoin, butyrate, octoic acid, acetic acid, octane, ether, 1,3-bis (1,1-dimethyl ethyl) benzene, n-hexanol, 2-heptanone, and hexanal were selected as the reasons for the differences in the volatile profile between these two groups.

## 4. Conclusions

The present study aimed to extend the shelf life of quark cheese trough DPCD processing. The optimal DPCD processing parameters (20 MPa, 45 min, 55 °C) were determined through orthogonal experiments. Temperature was considered as the most critical factor for DPCD treatment. Treatment with DPCD under the optimized conditions would cause a reduction of >7 Log CFU/g in the total bacterial count, and successfully hindered the growth of yeast during storage. The control became unsuitable for consumption at day 14 due to excessive yeast, whereas the cheese treated with DPCD was still within its shelf life at day 14. Cheese processed using DPCD displayed fewer alterations in pH, proteolysis and color, mainly due to the inactivation of enzymes as well as less microbial metabolism. Additionally, DPCD processing resulted in a more discontinuous network in the quark cheese, leading to a reduction in the *n*′ value and in rheological parameters and an extended serum phase in its microstructure. The 14-day storage period weakened the rheological properties for both cheeses and caused a more obvious moisture migration in the control. Additionally, DPCD could not significantly affect the volatile components of cheese overall at day 0. However, during the storage, DPCD-processed cheese had a better performance in terms of the retention of the volatile profile. This investigation is beneficial to the promotion of the practical application of DPCD processing technology in the food industry.

## Figures and Tables

**Figure 1 foods-11-02340-f001:**
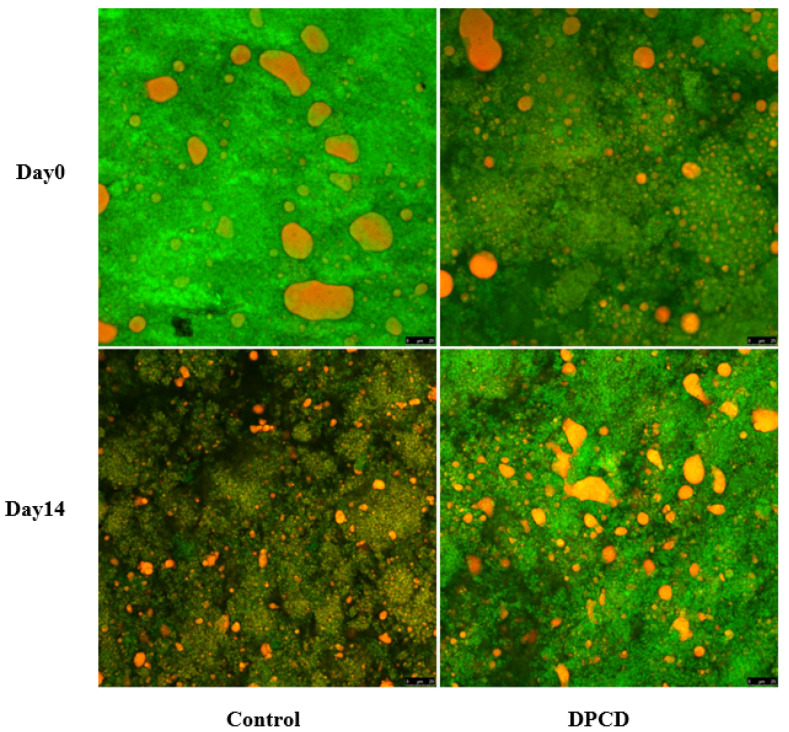
Microstructure of control and dense-phase carbon dioxide (DPCD)-processed quark cheese stored at 4 °C.

**Figure 2 foods-11-02340-f002:**
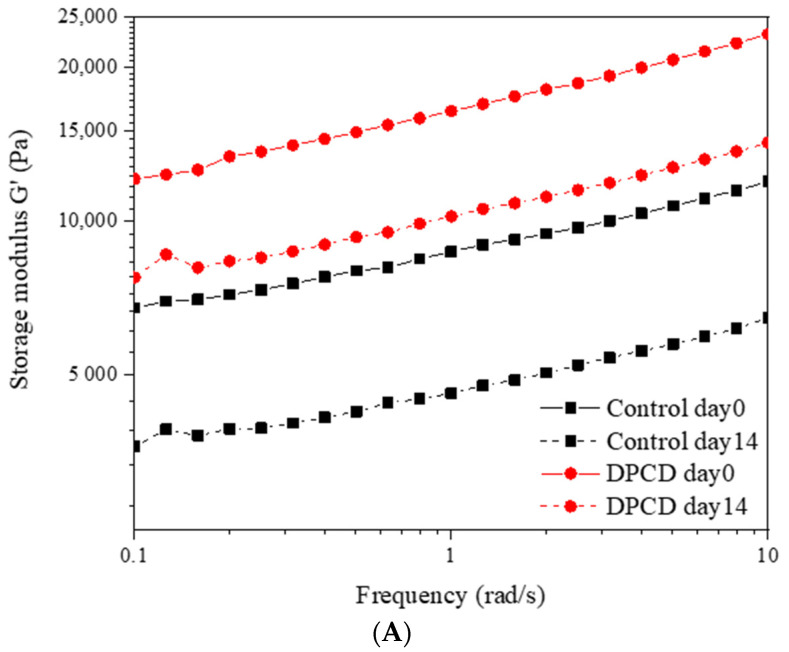
Storage modulus (*G*′) (**A**), loss modulus (*G*″) (**B**) and apparent viscosity (**C**) of control and dense-phase carbon dioxide (DPCD)-processed quark cheese stored at 4 °C.

**Table 1 foods-11-02340-t001:** Range analysis of orthogonal experiment for pressure, treatment time and temperature.

Number	A	B	C	TBC
	Pressure	Time	Temperature	(log CFU/g)
	(MPa)	(min)	(°C)	
1	10	25	35	7.26 ± 0.59 ^e^
2	10	35	45	6.39 ± 0.04 ^d^
3	10	45	55	2.94 ± 0.22 ^b^
4	20	25	45	5.59 ± 0.54 ^c^
5	20	35	55	2.70 ± 0.19 ^b^
6	20	45	35	7.23 ± 0.51 ^e^
7	30	25	55	2.83 ± 0.19 ^b^
8	30	35	35	7.84 ± 0.24 ^e^
9	30	45	45	1.71 ± 0.03 ^a^
K_1_	16.59	15.68	22.33	
K_2_	15.52	16.93	13.69	
K_3_	12.39	11.88	8.48	
k_1_	5.53	5.23	7.44	
k_2_	5.17	5.64	4.56	
k_3_	4.13	3.96	2.83	
R	1.4	1.69	4.62	

TBC represented the total bacterial count; initial TBC before DPCD treatment: 9.68 log CFU/g; data are mean values ± standard deviation (*n* = 3). The different lowercase letters in the same column indicate significant difference (*p* < 0.05).

**Table 2 foods-11-02340-t002:** Results of variance analysis of orthogonal experiment for pressure, time and temperature.

Source	SS	df	MS	F-Value	Sig.
model	733.211	7	104.744	114.967	0
A	8.565	2	4.282	4.7	0.23
B	11.984	2	5.992	6.577	0.007
C	95.541	2	27.77	54.432	0
Error	16.4	18	0.911		
Total	749.611	25			

R^2^ = 0.978; SS: sum of squares of deviations; df: degrees of freedom; MS: mean square; Sig.: significance; A: pressure; B: time; C: temperature.

**Table 3 foods-11-02340-t003:** Total bacterial count (TBC), total yeast and mold count (TYMC), pH, proteolysis activity and sensory evaluation of control and DPCD processed quark cheese stored at 4 °C.

Treatment		TBC	TYMC	pH	Proteolysis Activity(×10^−1^)	Sensory Evaluation
		(Log CFU/g)	(CFU/g)			
Control	Day 0	9.77 ± 0.09 ^Bc^	-	4.18 ± 0.02 ^Aa^	2.96 ± 0.08 ^Aa^	Qualified
	Day 7	8.80 ± 0.06 ^b^	-	4.23 ± 0.06 ^a^	3.23 ± 0.07 ^b^	Qualified
	Day 14	8.01 ± 0.13 ^a^	66.50 ± 23.33	4.42 ± 0.01 ^b^	3.48 ± 0.02 ^c^	-
DPCD	Day 0	2.44 ± 0.78 ^Aab^	-	4.30 ± 0.00 ^Ba^	2.87 ± 0.08 ^Aa^	Qualified
	Day 7	1.89 ± 0.09 ^a^	-	4.33 ± 0.01 ^b^	3.10 ± 0.04 ^b^	Qualified
	Day 14	2.57 ± 0.26 ^b^	3.50 ± 0.71	4.48 ± 0.00 ^c^	3.06 ± 0.01 ^b^	Qualified

Control represents the control without DPCD treatment. Data are mean values ± standard deviation (*n* = 3). The different lowercase letters indicate significant differences (*p* < 0.05) in the physicochemical properties with varying storage time; the different capital letters indicate significant differences (*p* < 0.05) in the physicochemical properties with or without DPCD treatment on day 0.

**Table 4 foods-11-02340-t004:** Color parameters including L*, a* and b* of control and dense-phase carbon dioxide (DPCD)-processed cheese stored at 4 °C.

Treatment		L*	a*	b*
	Day 0	94.13 ± 0.50 ^Aa^	−0.47 ± 0.12 ^Ba^	10.32 ± 0.24 ^Ab^
Control	Day 7	94.55 ± 0.19 ^b^	−0.25 ± −0.09 ^b^	10.26 ± 0.13 ^a^
	Day 14	94.84 ± 0.19 ^b^	0.13 ± 0.19 ^c^	9.41 ± 0.11 ^a^
	Day 0	94.77 ± 0.18 ^Aab^	−1.04 ± 0.05 ^Aa^	11.05 ± 0.12 ^Bb^
DPCD	Day 7	94.62 ± 0.23 ^a^	−0.95 ± 0.09 ^b^	10.37 ± 0.17 ^a^
	Day 14	94.99 ± 0.20 ^b^	−0.90 ± 0.06 ^b^	10.37 ± 0.37 ^a^

Control represents the control without DPCD treatment. Data are mean values ± standard deviation (*n* = 6). The different lowercase letters indicate significant differences (*p* < 0.05) in the color with varying storage time; the different capital letters indicate significant differences (*p* < 0.05) in the color with or without DPCD treatment on day 0.

**Table 5 foods-11-02340-t005:** Relaxation time (T_2i_) and relative content (*A*_2*i*_) of control and dense-phase carbon dioxide (DPCD)-processed cheese stored at 4 °C.

	Control		DPCD	
	Day 0	Day 14	Day 0	Day 14
T_21_	0.15 ± 0.00 ^Aa^	0.16 ± 0.00 ^b^	0.17 ± 0.01 ^Ba^	0.16 ± 0.01 ^a^
T_22_	1.15 ± 0.13 ^A^	-	1.72 ± 0.67 ^Aa^	2.80 ± 1.52 ^a^
T_23_	15.34 ± 0.00 ^Bb^	8.53 ± 2.41 ^a^	12.54 ± 0.72 ^Aa^	9.94 ± 4.40 ^a^
T_24_	95.00 ± 5.44 ^Ba^	88.19 ± 15.08 ^a^	84.07 ± 0.00 ^Aa^	84.07 ± 0.00 ^a^
T_25_	771.03 ± 259.86 ^Aa^	645.20 ± 146.53 ^a^	758.66 ± 172.30 ^Aa^	995.05 ± 56.96 ^a^
A_21_	12.45 ± 0.07 ^Aa^	13.25 ± 1.06 ^a^	12.05 ± 1.34 ^Aa^	14.70 ± 2.83 ^a^
A_22_	0.90 ± 0.00 ^A^	-	0.55 ± 0.21 ^Aa^	0.65 ± 0.49 ^a^
A_23_	3.85 ± 0.49 ^Aa^	2.60 ± 0.71 ^a^	3.60 ± 0.28 ^Aa^	3.35 ± 0.21 ^a^
A_24_	79.35 ± 1.06 ^Aa^	80.55 ± 2.05 ^a^	79.50 ± 0.42 ^Aa^	77.45 ± 2.05 ^a^
A_25_	3.45 ± 0.64 ^Aa^	3.55 ± 2.33 ^a^	4.35 ± 1.20 ^Aa^	3.85 ± 0.07 ^a^

Control represents the control without DPCD treatment. Data are mean values ± standard deviation (*n* = 2). The different lowercase letters indicate significant differences (*p* < 0.05) in T_2i_ and *A*_2*i*_ with varying storage time; the different capital letters indicate significant differences (*p* < 0.05) in T_2i_ and *A*_2*i*_ with or without DPCD treatment on day 0.

**Table 6 foods-11-02340-t006:** Quantity and relative contents of volatile compounds of control and dense-phase carbon dioxide (DPCD)-processed cheese stored at 4 °C.

Compound	Control				DPCD			
	Day 0		Day 14		Day 0		Day 14	
	*n*	RC (%)	*n*	RC (%)	*n*	RC (%)	*n*	RC (%)
Sulphocompounds	1	0.12	0	-	1	0.13	0	-
Aliphatic hydrocarbons	3	20.85	8	21.23	3	23.07	7	22.22
Heterocyclic compounds	2	2.38	2	1.61	1	1.68	2	1.90
Ethers	1	32.05	1	33.79	1	33.61	1	33.81
Acids	5	30.23	5	23.66	5	21.39	5	25.06
Alcohols	6	2.43	4	1.62	4	11.45	2	5.55
Aldehydes	1	0.09	1	0.07	2	0.34	2	1.08
Ketones	5	8.44	4	12.28	5	5.85	3	6.71
Aromatic hydrocarbons	7	3.42	5	4.39	6	2.47	4	2.93
Other compounds	0	-	1	1.35	0	-	1	3.05
Total	31		31		28		27	

Control represents the control without DPCD treatment; *n*: number of compounds; RC: relative content (%).

## Data Availability

The datasets used and/or analyzed during the current study are available from the corresponding author on request.

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
