# Peer review of "Quark Cheese Processed by Dense-Phase Carbon Dioxide: Shelf-Life Evaluation and Physiochemical, Rheological, Microstructural and Volatile Properties Assessment"

_foods, 2022, doi:10.3390/foods11152340_

Round 1

Reviewer 1 Report

General comments:

Improve English;

Do not start a sentence with an aberration or number, please correct it throughout the Manuscript;

The methodology must be improved;

State which controls were used in this trial;

Bacterial name – italic letters; ssp. lactis (small letter); Lactococcus lactis ssp. Butterfat what species is this?

There are no tables and pictures in the main Manuscript - it is difficult to do an adequate review

Reviewer 2 Report

Major comments

Table 6, it’s surprising to have high alcohol content (11.45%) at 0 day in DPCD sample and in table 3 TYMC count was not detected in the control and DPCD samples (L288-289) due to the strict hygienic conditions implemented by the milk company and your lab. Justify the source of high alcohol content in your samples in the light of no yeast culture added and bacterial cheese culture do not produce this amount of alcohol. Furthermore, justify the presence of TYMC after 14 days in control and DPCD.

Abstract:

L 25, L 32, Add cheese after quark

Add the aim of your study.

Introduction

L38, replace diary with dairy.

Manuscript must be revised by native English speaker. Do not start sentences with and (L133, L300, L350, L469, L535, L608, L618, L629).

L37-41, vague statement (rewrite).

L 48, L54, L79, L82, Add cheese after quark

Materials and Methods

L 93, provide pasteurization details because some types of pasteurization affect cheese coagulation.  

L 99, Add A volume of 350 ml……

L 105, replace sterilization with sterilized.

Rewrite the whole section (2.2)

L102, why rennet incubated for 10.5 h (very long time), rennet is an enzyme and takes about 30-45 min to coagulate milk.

L141-142, Qualified and non-qualified cheeses have to be classified by panelists. No enough details are provided in the methodology part.

L149, define L*, a*, b* values

Results and Discussion

L240, replace DCPD with DPCD.

Table 1, define A, B, C in the footnote.

L245-246, what about time?

Table 3, control has higher TBC and TYMC counts on the other hand lower alcohol content (table 6) compared to DPCD. Justify the source of high alcohol content in control and DPCD samples and discuss why it decreased after 14 days.

Rewrite the whole section (3.3)

L323-328, the control sample has the same increase in pH; discuss why?

L410-411, add a reference.

Table 6, discuss why acids increased in DPCD from 0 to 14 day; on the other hand, pH content in table 3 increased significantly after 14 days. This is not linked to L524-528, but discuss differences in acid content in DPCD from 0 to 14 days.

Table 6, discuss why acids decreased in control from 0 to 14 days.

Throughout the manuscript, the use of the English language needs to be improved. Native English speaker must revise it before resubmission. 

Round 2

Reviewer 2 Report

The authors have improved their manuscript and they have replied to most comments of the reviewers. However, I think authors still need to answer below comments.

You did not provide answer to my major comments even your response in point number 17 is not scientifically accepted because instruments in dairy factories are disinfected with alkaline (NaoH), it’s costly to disinfect with alcohol (tens of liters), and the last part of your response (alcohol was extremely volatile) contradict each other. In case alcohol volatile, it won’t appear in your tests (table 6). Please provide better response to this comment.

Point 19 and L403-405, provide reference for this justification.

Author Response

Response to Reviewer 2:

Point1: We are sorry that we might not explain it clearly. Firstly, as for the alcohol content, we did not attribute the existence of alcohol to the production process of milk from factories since there was no microbial contamination detected in cheese initially as we mentioned in results. However, it was likely we think attributed to the routine disinfection of our DPCD instrument in laboratory as various food materials was processed in it. But we believe that because of the volatile profile of alcohol, the amount of alcohol left in DPCD instrument was small that had little impact on cheese. However, the amount and content of volatile substance in quark cheese were in low level itself leading to a relatively large occupation of alcohol in GC-MS results. Secondly, in terms of the TYMC, it was reasonable that both control and DPCD processed cheeses were contaminated by fungi after 14 days and the control displayed greater contamination. We have also the TYMC of cheeses on day 21 (data was not shown in our manuscript) and the TYMC was 3.87±0.98 (×107) CFU/g in the control and 589.50±275.06 CFU/g in DPCD processed cheese, which could justify the results that DPCD inhibit fungal growth as we shown in manuscript. Evert-Arriagada et al. (2014) also reported that no fungus was detected on day 0 and it started appearing on day 7 in cheese samples. Additionally, HPP treated cheese showed less fungal contamination than the native cheese, which was consistent to our results.

[1] Evert-Arriagada, K.; Hernández-Herrero, M.; Guamis, B.; Trujillo, A. Commercial application of high-pressure processing for increasing starter-free fresh cheese shelf-life. LWT, 2014, 55(2), 498-505.

Point 2: We have complemented references for that.